# Investigation of Changes to Triaxial Shear Strength Parameters and Microstructure of Yili Loess with Drying–Wetting Cycles

**DOI:** 10.3390/ma15010255

**Published:** 2021-12-29

**Authors:** Ruihua Hao, Zizhao Zhang, Zezhou Guo, Xuebang Huang, Qianli Lv, Jiahao Wang, Tianchao Liu

**Affiliations:** 1School of Geological and Mining Engineering, Xinjiang University, Urumqi 830017, China; 18220186964@163.com (R.H.); xiangerji427@163.com (Z.G.); hxb0714@163.com (X.H.); 13579843644@163.com (Q.L.); WJH1254051722@163.com (J.W.); 2State Key Laboratory for Geomechanics and Deep Underground Engineering, Xinjiang University, Urumqi 830017, China; 3The First Regional Geological Survey Brigade, Xinjiang Bureau of Geo-Exploration and Mineral Development, Urumqi 830011, China; ltc0524@126.com

**Keywords:** drying–wetting cycles, yili loess, triaxial shearing parameters, nuclear magnetic resonance (NMR), scanning electron microscopy (SEM)

## Abstract

This research examined the drying–wetting cycles induced changes in undrained triaxial shear strength parameters and microstructural changes of Yili loess. The drying–wetting cycles were selected as 0, 1, 3, 5, 10, 20 and 30. Then, we collected Yili loess samples and performed unconsolidated-undrained (U-U) triaxial shearing tests to ascertain the variation in shear strength parameters with drying–wetting cycles. Additionally, we investigated the microstructural changes of Yili loess samples under drying–wetting cycles simultaneously via nuclear magnetic resonance (NMR) and scanning electron electroscopy (SEM). Finally, we established a grey correlation model between shear strength and microstructural parameters. Under U-U conditions, the prime finding was that the loess’s shear strength parameters changed overall after drying–wetting cycles; in particular, the internal friction angle φ dropped significantly while the cohesion c changed only slightly during cycles. For all the cycles, the first cycle gave the highest change. Soil morphology deterioration was evident at the initial stage of cycles. During the entire drying–wetting cyclic process, pore size distribution showed progressive variance from two-peak to a single-peak pattern, while both porosity and the fractal dimension of pores increased gradually towards stability. Soil particle morphology became slowly simple and reached the equilibrium state after 20 drying–wetting cycles. Under cyclic drying–wetting stress, the shear strength parameter changes were significantly correlated to microstructural modifications. This investigation was related to loess in the westerly region. The findings were expected to provide new insight into establishment of the connection between microstructure and macro stress–strain state of loess. To some extent, it provided a theoretical basis for the prevention and control of loess engineering geological disasters in Yili, Xinjiang and other areas with similar climate and soil types.

## 1. Introduction

Yili basin is located in Yili, Xinjiang, China, which is adjacent to the western border of the Republic of Kazakhstan. It is a critical Eolian loess region in China (typical westerly loess) [1]. Yili loess shows the banded distribution in terraces along many rivers, including Yili River, Kongnaisi River, Tekesi River, Kashi River, Qialun River, and Qilike River. It is characterized by low hilly areas, piedmont slopes, and the marginal zones of the desert, with varying thickness from several to nearly 100 m [2,3,4]. The Yili area has abundant precipitation events and has the characteristics of a humid continental mesotemperate climate. The precipitation in this area is mainly concentrated from April to July, accounting for about 51.5–80% of the whole year. During the same periods, the annual average evaporation is 1301.5 mm, which is 2.7 times of the precipitation. The loess in Yili River Valley is affected by repeated rainfall humidification and evaporation dehumidification. Therefore, it is of great significance to study the impact of drying–wetting cycles on Yili loess, which could provide a basis for engineering construction and geological disaster prevention in the Yili area as well as other regions with the same climate and soil types.

A substantial amount of research has been carried out to study the mechanical properties of various soils. Chindaprasirt et al. [5] estimated the elastic modulus of loess and laterite by laboratory test. Xiao et al. [6] analyzed the fractal dimension and its variation law of undisturbed loess and compacted loess. Xu et al. [7] found that the infiltration of inorganic salt solution into remolded loess will deteriorate its engineering properties. Gao et al. [8] conducted a detailed experimental study on particle breakage of loess, silty loess and cohesive loess. It was found that there was particle breakage in loess, and the initial fracture stress of silty soil and silty soil was less than 0.5 MPa for cohesive loess and about 2MPa for sandy loess. Somavilla et al. [9] found that wetting and drying processes can change particle rearrangement, cementation, and related properties in soil. Salih et al. [10] studied the characteristics of red mud waste modified by desulfurization gypsum-fly ash. It was found that with the increase of drying–wetting cycles, the characteristics and structure of the mixture samples changed significantly, the porosity of the sample structure also increased, and the density and unconfined pressure resistance of the sample structure decreased. Tarantino and Mountassir [11] proposed that shear strength can be conservatively characterized by constant water content test without facility control/monitoring suction, and water retention can be successfully predicted by single water retention measurement. Sorbino and Nicotera [12] described the process of rainfall induced landslide, and briefly reviewed some observations on its modeling. De Bono and McDowell [13] found through the stress path test that the shear yield surface depends on the stress history, while the compression yield surface is isotropic. Miščević [14] found that the deterioration of particles in marl caused by the wetting and drying process led to additional settlement of embankment made of crushed soft rock particles.

Currently, scholars mainly focus on the changes of surface crack [15], soil shear strength [16,17,18], soil permeability under drying–wetting cyclic action [19,20], and the induced structural damages on the soil [21,22]. In addition, Fuyang Cheng et al. [23] considered that the number of drying–wetting cycles imposed a more significant effect than the amplitude of cycles and concluded that the change of shear properties essentially lies in the interaction between soil and water. Hao et al. [24] considered that the deterioration degree of triaxial shear strength caused by drying–wetting cycles can be evaluated by the instantaneous water content of loess after compaction. In addition, by combining macro-mechanical properties and microstructural analysis, many scholars concluded that the drying–wetting cycle leads to the agglomeration and dispersion of soil particles and the change of soil pores, thereby leading to structural loss and soil intensity attenuation [25,26,27,28,29]. However, the changes in physical properties and microstructure of loess samples from Yili Valley under drying–wetting cycles have not been well understood. Therefore, this study focused on Yili loess, selected the number of drying–wetting cycles as the control variable, employed triaxial shear, NMR, and SEM tests to analyze the soil’s shear properties and micro-structural properties, and reveals the micro process of shear strength change on the macro-level under the condition of drying–wetting cycles.

Our results demonstrated that drying–wetting cyclic action significantly influenced Yili loess shear properties and the corresponding microstructural characteristics. At the macroscopic level, Yili loess demonstrated shear strength, cohesion, and internal friction angle drop; micro-structural damages occurred inside the Yili loess at the microscopic level. This study explored the strength and microstructure changes of loess from westerly areas under drying–wetting cycles condition, which provided a theoretical basis for the prevention and control of loess engineering geological disasters in Yili, Xinjiang.

## 2. Materials and Methods

### 2.1. Materials

We collected the loess samples from specific areas around Alemale Town, Xinyuan County, Kazakh Autonomous Prefecture, and Yili (Figure 1). Yili loess is composed of a large proportion of silt and a certain amount of fine sand. Overall, the particle size of Yili loess exceeds that of the Loess Plateau, while the sorting property is inferior to that of the Loess Plateau. Yili loess is mainly composed of quartz, feldspar, and carbonate minerals in mineral composition, with specific amounts of chlorite and mica [30]. Other scholars have studied the deformation, strength, collapsibility, and water characteristics of Yili loess. For example, Wei et al. [31] found that the tensile strength of Yili loess increases with the decrease of water content. Yuan et al. [32] consider that the tensile strength of Yili loess has a negative exponential relationship with water content. Wang et al. [33] consider that the confining compressive strain gradually increases with the increase of dry–wet cycles, and the initial compaction has a significant impact on the compression deformation of loess. Aijun et al. [34] found that there is a linear relationship between the total suction and solution concentration of Yili loess, and the osmotic suction does not exceed 25,000 kpa. Wen et al. [35] consider that significant hysteresis of SWRCs was observed under the first drying and wetting cycle. And the hysteresis effect decreased as the number of drying and wetting cycles increased.

The sampling depth of the test in this study was set as 2 m. After sampling, the natural moisture content was measured by drying method immediately. The indoor test was started to measure the basic physical parameters of loess in this area after the samples were taken back: the saturated moisture content was measured by the drying method, the optimal moisture content, and maximum dry density were obtained by the compaction test, and the porosity and void ratio were obtained by nuclear magnetic resonance. The results are shown in Table 1. Mineral composition analysis (XRD) was carried out after grinding the soil, and the results are shown in Table 2. In addition, Lisi et al. [36] and WANG et al. [37] analyzed the soluble salt content of Yili loess. The analysis results are shown in Table 3. In this paper, the change of soluble salt during circulation and its impact on soil strength are not considered, therefore the relevant tests are not carried out.

### 2.2. Methods

#### 2.2.1. Drying–Wetting Cyclic Process

After air drying, according to Standard for Geotechnical Testing Method [38], Yili loess was crushed and sifted by the sieve with a diameter of 2 mm to prevent gravel from acting as a skeleton in the soil and affecting the test results (strength and deformation characteristics). The sample was prepared with a moisture content of 17.4% (optimum moisture content measured by compaction test); next, the prepared sample was in the moisturizing dish, sealed with a preservative film. After 24 h of storage, it is assumed that the water in the soil reaches a state of uniform distribution. Finally, after compaction, the cylindrical samples with sizes of Φ 39.1 mm × 80.0 mm and Φ 50.0 mm × 25.0 mm were prepared for triaxial shear testing and microstructural analysis (as shown in Figure 2). The sample density was 1.86 g/cm^3^ (the maximum dry density). Table 4 lists the number of samples for different tests.

This study set the number of drying–wetting cycles as 0, 1, 3, 5, 10, 20, and 30, respectively. During the cyclic process, the initial moisture content of the prepared loess sample was 17.4%. Under natural conditions, Yili loess is wetted to saturation during the rainfall process; water in the soil 2 m below the surface did not evaporate thoroughly, and soil cannot reach a dry state during the evaporation process. Therefore, the lower moisture content limit in the drying–wetting cycles was 10%, and the upper limit (i.e., the saturated moisture content measured by drying method) was 24.57%. Thus, during the single drying–wetting cycle period, the loess sample showed a cycle of initial moisture content-saturated moisture content-lower limit moisture content-initial moisture content and underwent three steps: humidifying, dehumidifying, and humidifying. Figure 3 illustrates the drying–wetting cyclic process.

During the humidifying process, the loess sample became saturated through soaking. We wrapped a layer of preservative film on the cylinder of loess sample, and placed filter paper on the upper and lower surfaces. Then the polyvinyl chloride (PVC) pipe was laterally restrained, and the porous permeable stone was placed on the filter paper on the upper and lower surfaces as the axial control to minimize the interference of the sample. After that, the sample was wrapped with a layer of preservative film, and holes with a diameter of 0.5 mm was cut on the preservative film on the upper and lower surfaces of the sample. The usage of PVC pipe is shown in Figure 4. After soaking in distilled water for 12 h, the sample achieved the approximate saturated state and ended the humidifying process, and it was placed in the moisturizing dish for 12 h for uniform dispersion of water.

In regard to the dehumidifying process, the sample was dried by drying oven. In the beginning, we removed the preservative film on the outermost layer of the sample and the permeable stone on the upper and lower surfaces. Then the samples were placed with the PVC lateral restraint horizontally at 60 °C for slow evaporation to guarantee only the upper and lower surfaces have evaporation. The sample was weighed hourly.

Before the test, the mass measured quality of the *PVC* pipe used. And then the instantaneous moisture content was calculated based on the mass measured per hour. The calculation formula is shown as follow:(1)$$\omega_{h}=\frac{m_{h}-v_{sample}\cdot \rho_{dry}-m_{PVC}}{v_{sample}\cdot \rho_{dry}}$$

*m_h_* is the mass measured per hour during drying, and *ω_h_* is the instantaneous moisture content.

Then, the dehumidifying process ended when the instantaneous moisture content achieved the preset moisture content. The sample was kept undisturbed in the moisturizing dish, achieving uniform water distribution, and completing the dehumidifying process.

To meet the requirements of the triaxial shear test, the size of the sample during preparation was Φ 39.1 mm × 80.0 mm. In the process of soil humidification, due to the existence of lateral and axial restriction, the volume of the sample will not be changed. In the process of dehumidification, due to the removal of axial restriction for water evaporation, the sample has slight expansion in the axial direction. Before the triaxial test, the expansion part of the sample shall be removed according to the requirements of the test instrument for the size of the sample.

#### 2.2.2. Triaxial Shear Test

After the preset number of drying–wetting cycles, the triaxial shear test was performed on the sample. The samples were placed in seven groups according to the number of drying–wetting cycles (0, 1, 3, 5, 10, 20, and 30). Each group contained four samples. The drying–wetting cycles cause the soil structure to become loose while the consolidation effect will offset this influence, therefore, the unconsolidated and undrained (U-U) shear test was selected. The experimental instrument is the Model TFB-1 Non-saturated Soil Stress-Strain Controlled Triaxial Test Apparatus. The ambient temperature was set as 20 °C. According to Standard for Geotechnical Testing Method, the surrounding pressure shall be adapted to the minimum principal stress σ_3_ of the soil, so the design confining pressure was 50, 100, 200, and 300 kPa. The shear strain rate shall be 0.5%/min–1.0%/min, the sample height shall be 80 mm, and the calculated rate shall be 0.4 mm/min–0.8 mm/min. The final shear rate is determined as 0.5 mm/min.

#### 2.2.3. Microstructure Analysis

After completing the planned drying–wetting cycles, SEM and NMR was used for microstructure analysis. SEM employs an electron beam for grating-shaped scanning on the sample surface to obtain SEM images of the sample morphology. The SEM images of the samples were magnified to 500, 1200, 1500, 2000, and 2500 times, respectively. As regards NMR, a group of standard samples with known water content were tested to fit a curve between the amount of water and NMR signal. Then, the NMR signals of samples after different drying–wetting cycles were tested. The measured signals were brought into the curve equation to calculate the amount of water in the sample to obtain the porosity. The samples were tested only before the shear test to obtain the change of soil porosity under the action of drying–wetting cycles.

## 3. Results

### 3.1. Analysis of Triaxial Shear Test Results

#### 3.1.1. Stress–Strain Relationship

We obtained the stress–strain relationship and the variation characteristics of shear strength of the Yili loess sample under drying–wetting cyclic action by analyzing the triaxial shear test data.

The relation curves between principal stress different and axial strain of Yili loess samples after varying numbers of drying–wetting cycles at a confining pressure of 300 kPa (Figure 5). Under unconsolidated-undrained (U-U) condition, the confining pressure remained unchanged, and the relation curve after no drying–wetting cycle was the highest, suggesting soil deterioration after drying–wetting cycles, thereby prompting the decline in soil strength. As the number of drying–wetting cycles increased, the relation curves fluctuated, but peak value dropped, indicating strain softening.

The variations of peak principal stress difference and peak axial strain of Yili loess samples with different numbers of drying–wetting cycles (Figure 6). As the number of drying–wetting cycles increased, peak principal stress difference decreased gradually and shifted to the left, which indicated that soil strength dropped with drying–wetting cycles at a lower strain level. Moreover, the peak axial strain corresponding to peak principal stress difference overall fell, suggesting weakened shear failure resistance ability of Yili loess; the sample reached the peak under minor stress, resulting in shear failure.

#### 3.1.2. Variation Characteristics of Shear Strength Parameters

The test data were analyzed by fitting major and minor principal stress (σ_1_ and σ_3_) to obtain shear strength parameters of Yili loess samples after different drying–wetting cycles; the results are shown in Table 5.

The variations of two shear strength parameters of Yili loess samples (internal friction angle φ and cohesion C) with the number of drying–wetting cycles (Figure 7). Under unconsolidated-undrained (U-U) conditions, the internal friction angle of the sample dropped in a fluctuant pattern after drying–wetting cycles; overall, the effects of drying–wetting cycles at the beginning exceeded that in the later period. The soil’s cohesion also varied slightly during drying–wetting cycles in an irregular pattern. It was thus conclusively demonstrated that the soil’s cohesion was affected somewhat by the cyclic drying-wetting action, and the change of soils strength was mainly due to the reduction of internal friction angle.

#### 3.1.3. Variation Characteristics of Shear Strength Parameters

Given the determined shear strength parameters (c and φ), the shear strengths of Yili loess samples after different numbers of drying–wetting cycles were from the Mohr–Coulomb equation (Equation (2)):(2)τ=c+σtanφ

We list the calculated results in Table 6.

Figure 8 shows the relationship between different drying–wetting cycles and shear strength. Obviously, the higher the confining pressure, the greater the shear strength. The shear strength shows a similar variation law under each confining pressure. Under unconsolidated and undrained conditions, the shear strength of the sample without drying–wetting cycle (*N* = 0) is the largest and far greater than the value subjected to dry wet cycle. After the initial drying–wetting cycle, the shear strength of the sample decreases significantly and gives a lower value. This is because the original structure of the soil is broken and the internal structure of the soil changes significantly after humidification and dehumidification. With the increase of the number of drying–wetting cycles, the shear strength of the sample shows a fluctuation, and the variation range is less than that of the first cycle. Owing to the internal structure of soil is gradually stable after many dry wet cycles, when the number of drying–wetting cycles is less than 10, the shear strength of the sample changes sharply; when it exceeds 10, the change of shear strength tends to be relatively flat.

### 3.2. Analysis of Microstructural Test Results

#### 3.2.1. Analysis of NMR Test Results

Before the test, a group of standard samples with known water content were tested to fit a curve between the amount of water and NMR signal. Then, the NMR signals of samples after different drying–wetting cycles (0, 1, 3, 5, 10, 20, and 30) were tested. The measured signals were brought into the curve equation to calculate the amount of water in the sample to obtain the porosity. The results are shown in Table 7.

The porosity of the sample was enhanced after cyclic drying–wetting action. Notably, after the first drying–wetting cycle and three cycles, porosity changed significantly. As the number of drying–wetting cycles increased, the irregularity of porosity decreased gradually. This migration of water in soil when the compacted soil underwent the first drying–wetting cycle led to the migration of fine particles in soil; accordingly, pores in the soil became connected to form pore water transport channels. The above phenomenon was quite apparent at the beginning of drying–wetting cycles, causing a significant porosity change. In the middle and later phases of drying–wetting cycles, effective water transfer channels had already formed in the soil, and the migration effect of soil particles by water was weak. At that moment, particles in soil samples were stable, and the porosity change dropped gradually and stabilized.

Pore size distribution of loess samples after different dry–wet cycles are shown in Figure 9. Overall, the pores in the soil samples were mainly 0.01 μm~100 μm in diameter. Under drying–wetting cyclic action, pore size distribution curves gradually moved towards the right, and an increased number of pores with a diameter of 10 μm appeared in the samples. It suggested that the number of tiny pores gradually dropped while large and medium pores increased, facilitating failure surface formation around water transfer channels and large pores. Sliding surfaces were easily formed after the interconnection of water transfer channels and pores, reducing the soil’s shear strength, and causing soil deterioration. As the shear strength parameter, the internal friction angle reflects the friction between soil particles resisting the deformation in shear change, mainly sliding friction and occlusal friction among soil particles. Because of the irregularity of soil particles, the soil’s internal friction angle was remarkably in its original state; sliding surfaces were easily formed after drying–wetting cycles. The frictional force among soil particles changed from occlusal to sliding friction, while sliding friction was lower than occlusal friction. Therefore, the soil’s internal friction angle dropped by a certain degree after drying–wetting cycles.

The pore size distribution curve of the sample without drying–wetting cycles shows two peaks at the pore diameters of 1 μm and 20 μm, respectively. However, as the number of drying–wetting cycles increased, the pore size distribution pattern changed from two-peak to single-peak form, and the peak only existed at a pore size of 1μm. Thus, pore size distribution patterns changed under drying–wetting cycles, i.e., the diameters of pores in loess samples became more uniform.

#### 3.2.2. Analysis of SEM Test Results

After several drying–wetting cycles, the loess samples were observed in a scanning electron microscope at different amplification factors (500, 1200, 150, 2000, and 2500). The SEM images at an amplification factor of 500 were blurry. As the amplification factor exceeded 1200, fewer soil particles were observed, and many pores occupied the photos. Therefore, the amplification images of Yili loess samples after different drying–wetting cycles at a factor of 1200 were selected for further analysis, as shown in Figure 10.

As shown in Figure 10, soil particles became fragmented after drying–wetting cycles. Large particles were crushed to form smaller particles and aggregates. Large pores in soil were progressively filled by aggregates of soil particles. As a result, the number of large particles fell while the number of small particles during the aggregation process also decreased, but not obviously. Finally, the number of medium pores increased steadily during wetting-drying cycles because of large particles’ fragmentation and aggregation of small particles.

The SEM images were digitalized with Image-Pro Plus (IPP, a digital image processing software, Version 6.0, Media Cybernetics, Inc., Rockville, MD, USA) to extract related soil porosity parameters, including pore area, pore diameter, and the fractal dimension of pores. Then, each image was processed and removed in the same way to ensure the slightest artificial difference.

The statistics of pore area, pore diameter, and pore fractal dimension were obtained via digital processing of SEM images; the details are shown in Table 8.

The relationship curves between the calculated micro-structural parameters and the number of drying–wetting cycles are shown in Figure 11.

It was evident that the soil’s pore area, maximum pore diameter, minimum pore diameter, and mean pore diameter showed similar variations. Overall, as the number of drying–wetting cycles increased, all these parameters first decreased, rose, fell, and finally stabilized. Under drying–wetting cyclic action, soil first expanded when achieving a saturated state, and pores among the soil particles decreased; soil particles shrunk during the dehydration process, accompanied by increased pore size. In addition, soil particles became rearranged during drying–wetting cyclic. After the first drying–wetting cycle finished, the distribution of soil particles was nonuniform, and some large pores were segmented into tiny pores, increasing the specific surface area of soil particles and a decreased pore area. Meanwhile, both maximum and minimum pore diameters dropped. After five drying–wetting cycles, some soil particles agglomerated during the dehydration process, and large aggregates were formed in the soil, resulting in a small specific surface area of soil particles and increased pore area. Accordingly, pores among soil particles increased, accompanied by increasing maximum, minimum, and mean pore diameters. As the number of drying–wetting cycles further increased, the sample’s distribution of soil particles and pores showed similar structure and pattern, and the variation of pore area and pore diameter tended to stabilize.

During drying–wetting cycles, the fractal dimension of pores in the sample increased overall. A pronounced increase of the fractal pore dimension of pores existed at the beginning of drying–wetting cycles. The fractal dimension tended to stabilize after 20 drying–wetting cycles, suggesting increased simple soil particle patterns during the cycles. The first drying–wetting cycle imposed the most significant effects, while soil arrived at a new equilibrium after 20 drying–wetting cycles.

### 3.3. Grey Correlation Analysis between Shear Strength Parameters and Microstructural Parameters

Based on U-U triaxial shearing tests, we found that soil’s undrained shear strength parameters changed with cyclic drying–wetting action. The microstructural analysis also indicated the changes in soil’s microstructure during these cycles. Both types of parameters showed similar variations. Accordingly, the micro process of shear strength change on the macro-level is explored through correlation analysis between shear strength and microstructural parameters.

We established the present model via grey correlation, and by setting both reference and comparative sequences, the correlation degree was determined with the similarity degree of their geometrical shapes. And, a greater geometrical similarity degree is indicative of a higher correlation degree, corresponding to a more significant effect of evaluation objects on reference objects. Thus, the products on reference objects were evaluated.

According to quantitative analysis results of micro-structure, mean pore diameter and the fractal dimension of pores were adopted as comparative sequence Xi(k). At the same time, the shear strength parameters of Yili loess samples after different numbers of drying–wetting cycles were set as reference sequences Yi(k). Table 9 lists the original data.

On account of different dimensions and units of various parameters, non-dimensional indices in reference and comparative sequences were obtained to establish certain relations. All original data were initialized via averaging, as the dimensionless processing results are listed in Table 10.

By adopting Xi(k) and Yi(k) as characteristic sequence and mother sequence, respectively, the correlation coefficients and the correlation degrees were calculated based on initialization processing results, as the data shows in the Table 11 and Table 12. A more significant correlation degree suggests a more favorable correlation. Generally, a correlation degree of over 0.6 meant correlation, and a correlation below 0.6 suggested poor correlation; these two indices were almost uncorrelated if the correlation was below 0.5.

According to the calculated correlation coefficients and degrees between shear strength parameters and microstructural parameters of Yili loess samples after different drying–wetting cycles, we found that: (a) mean pore diameter imposed almost no effect on internal friction angle; (b) as the number of drying–wetting cycles increased, the correlation coefficient increased; mean pore diameter began to be correlated with internal friction angle; (c) mean pore diameter was well correlated with cohesion under cyclic drying–wetting action; (d) the fractal dimension of pores were highly correlated with shear strength parameters during the whole cyclic process. Overall, the correlation between microstructural parameters (mean pore diameter and the fractal dimension of pores) and internal friction angle was weaker than the correlation between mean pore diameter and cohesion. Mean pore diameter and the fractal dimension of pores imposed more significant effects on the cohesion parameter.

Internal friction angle among shear strength indices is mainly sourced from the friction among soil particles, while mean pore diameter cannot reflect soil particle morphology, suggesting a weak correlation between internal friction angle and mean pore diameter. The fractal dimension of pores demonstrates the degree of irregularity of pores, reflecting the magnitude of friction among soil particles to a certain degree. Accordingly, the correlation between the fractal dimension of pores and the internal friction angle is strong. Cohesive force comes from inter-attraction among soil particles. The larger mean pore diameter and fractal dimension of pores indicate a tighter arrangement of soil particles, more surface morphology, and a more potent cohesion parameter. By contrast, the mean pore diameter and the fractal dimension of pores correlated well with the cohesion parameter.

## 4. Discussion

Current research scholars have focused their research emphasis on loess in westerly wind areas regarding the effects of dry density, moisture content, the content of soluble salt on mechanical properties, and collapsibility. They pointed out that the drying density, moisture content, and soluble salt content of the Yili loess sample correlated with shear strength, tensile strength, modulus of compressibility, and collapsibility. The existing studies regarding the change of soil mechanical properties under drying–wetting cyclic action have mainly focused on red soil in southwestern China and loess on Loess Plateau. Yili in Xinjiang Autonomous Region featured abundant precipitation and intensive evaporation. Loess in the westerly region also underwent repeated rainfall-induced wetting and evaporation-induced dehumidifying cycles, with intense cyclic drying–wetting action. This study combined macro-mechanical tests and microstructural analysis for exploring the influencing mechanisms of drying–wetting cycles on the mechanical properties of Yili loess. We found that the degradation of Yili loess strength induced by drying–wetting cycles changed the soil particles and pore distribution. The present research contents can contribute to exploring the deformation of Yili loess under drying–wetting cyclic action to address the geological hazards of loess landslides widely distributed in Yili.

We performed unconsolidated-undrained (U-U) triaxial shearing tests, NMR, SEM under drying–wetting cycles, and figured that: the redistribution of soil particles and pores between particles caused by repeated humidification and dehumidification is the direct reason for the change of soil strength and structure. Soil particles constantly underwent crushing and re-arrangement during the cyclic humidifying and dehumidifying process. After repeated drying–wetting cycles, both size and shape of soil particles became more uniform; accordingly, large and small pores decreased gradually in number, and simultaneously, the increase of medium pores reduced occlusal friction among soil particles to a certain degree, accompanied the declines in internal friction angle and soil’s shear strength. The change gradually weakened after 20 drying–wetting cycles; afterward, soil varied slightly under drying–wetting cyclic action and reached an equilibrium state.

In this study, remolded soil was used for laboratory tests, and soil samples are prepared with the maximum dry density and optimal moisture content. The basic physical parameters of the samples are relatively single, and there are great differences in structure between remolded soil and undisturbed soil; therefore, it is necessary to further consider the soil microstructure under natural conditions. Furthermore, tests should be taken under different cyclic drying–wetting conditions (to be specific, different amplitudes of cycles and drying–wetting cyclic paths) to explore the variation rules of mechanical properties of Yili loess in a natural state under other restricted conditions. In addition, more microstructural parameters should be selected to comprehensively explain the variation process of macro-mechanical properties on the micro-level.

In this paper, the undrained-unconsolidated shear test is used to investigate the shear strength under different drying–wetting cycles; however, the selection of drying–wetting cycles is still insufficient. It does not fully reflect the change law of the sample under the continuous drying–wetting cycle (i.e., when *N* is greater than 30); therefore, the results obtained in this study have certain limitations. In addition, the samples in different stages of drying–wetting cycles are not tested, which cannot reflect the structural changes of the samples in a single cycle. This paper only provides the change law based on the results of drying–wetting cycle. Further, when SEM and NMR technology are employed, more extensive sample results are easier to be accepted. Samples in different states and stages should be analyzed and explored in more detail in future work.

## 5. Conclusions

Yili loess samples show reduced strength and structural changes under drying–wetting cycles in the following aspects:(1)Under unconsolidated-undrained (U-U) conditions, shear strength the soil fell after drying–wetting cycles, especially at the beginning of cycles, which finally tended to be stable with the increasing drying–wetting cycles.(2)Shear strength parameter φ of the soil was significantly affected under drying–wetting cyclic action, which first decreased and then tended to be stable during the whole cycle. On the other hand, the shear strength parameter c was slightly affected and showed irregular variations over the entire cycle.(3)The overall porosity of the soil increased with the number of drying–wetting cycles. The porosity increase became pronounced in the first drying–wetting cycle. During the cycle, the soil’s pore size distribution changed from two-peak to single-peak pattern.(4)During the repeated wetting and dehumidifying process, the change of soil particles induced regular variation of pore size distribution. Pore area, maximum pore diameter, minimum pore diameter, and mean pore diameter all dropped at the beginning of drying–wetting cycles. After five drying–wetting cycles, pore size increased slightly and finally tended to be stable with the increasing number of drying–wetting cycles.(5)The fractal dimension of pores increased under drying–wetting cyclic action and tended to be stable after approximately 20 drying–wetting cycles. Soil particle morphology became gradually simple during the drying–wetting cycles, which also remained stable after 20 cycles.(6)The grey correlation model between shear strength and microstructural parameters showed that pore diameter and pore fractal dimension correlated well with cohesion, but pore diameter correlated poorly with internal friction angle.

## Figures and Tables

**Figure 1 materials-15-00255-f001:**
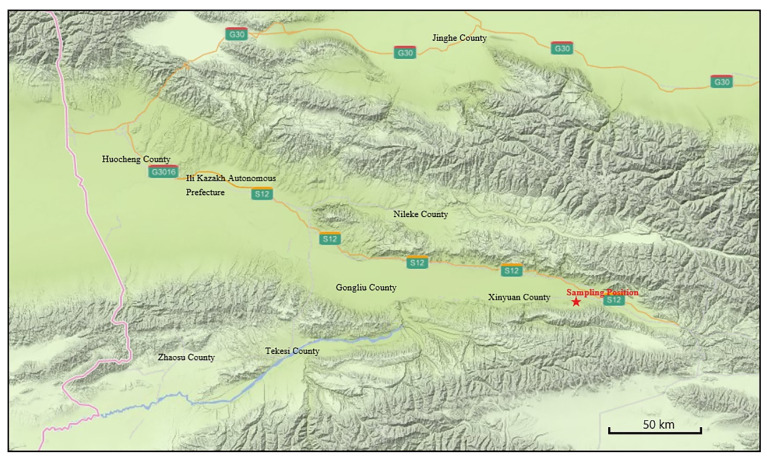
Illustration of sampling positions of Yili loess.

**Figure 2 materials-15-00255-f002:**
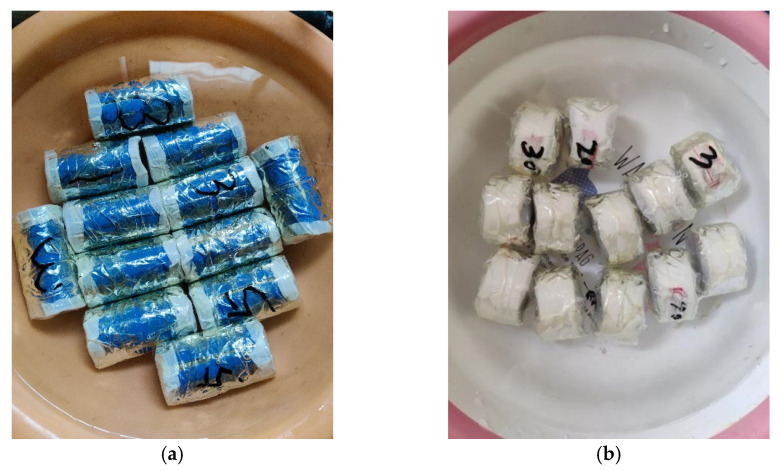
Pictures of the test samples (**a**) Samples for triaxial shear test, (**b**) Samples for microstructural analysis.

**Figure 3 materials-15-00255-f003:**
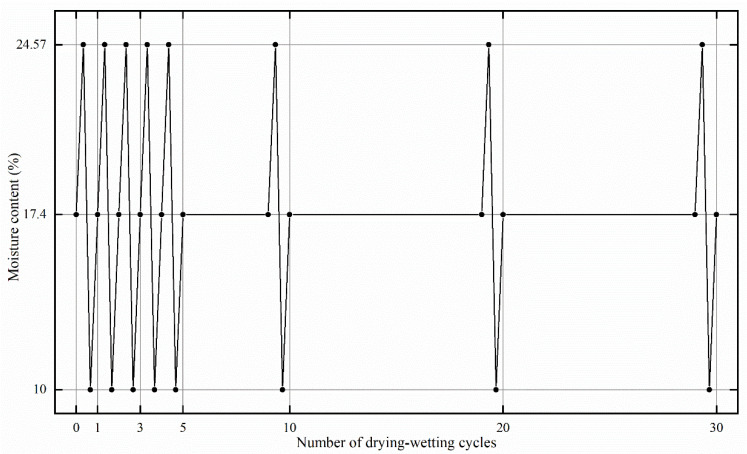
Illustration of drying–wetting cyclic process.

**Figure 4 materials-15-00255-f004:**
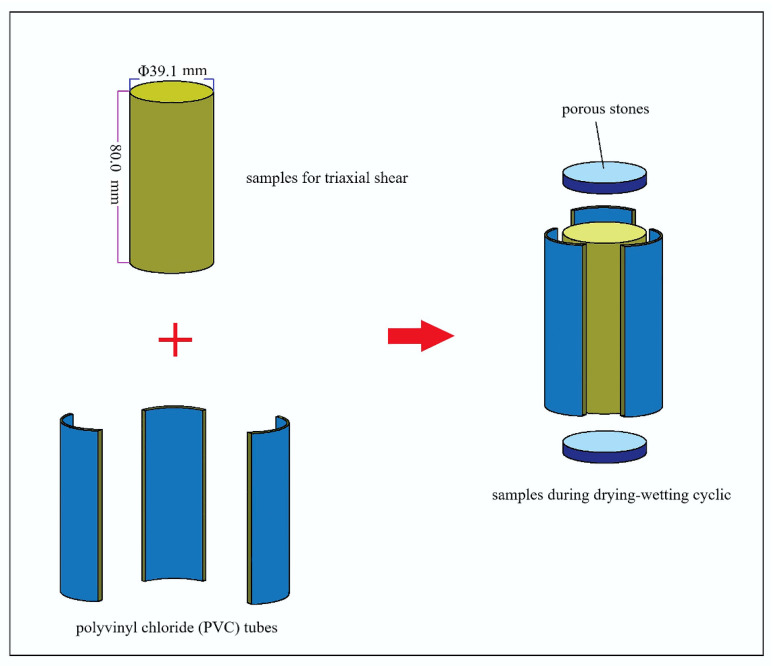
Illustration of usage of PVC pipe during drying–wetting cyclic process.

**Figure 5 materials-15-00255-f005:**
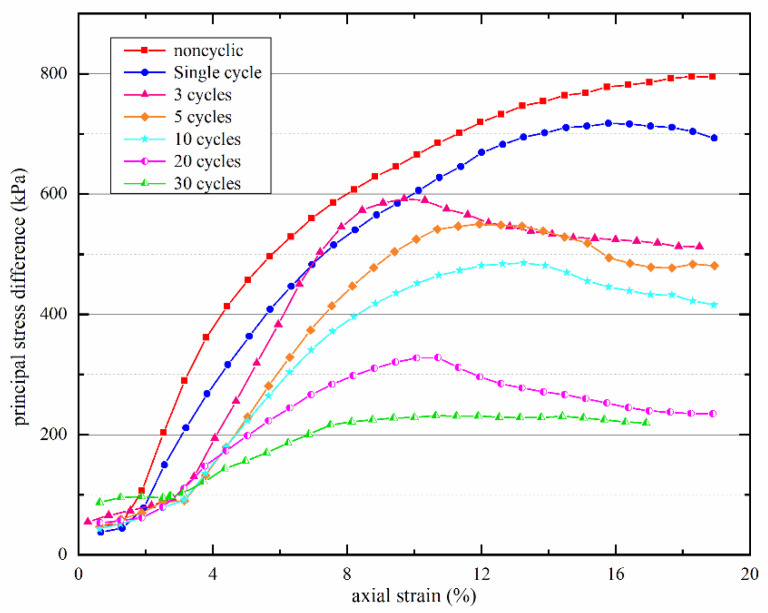
The stress–strain curves of Yili loess samples after varying numbers of drying–wetting cycles.

**Figure 6 materials-15-00255-f006:**
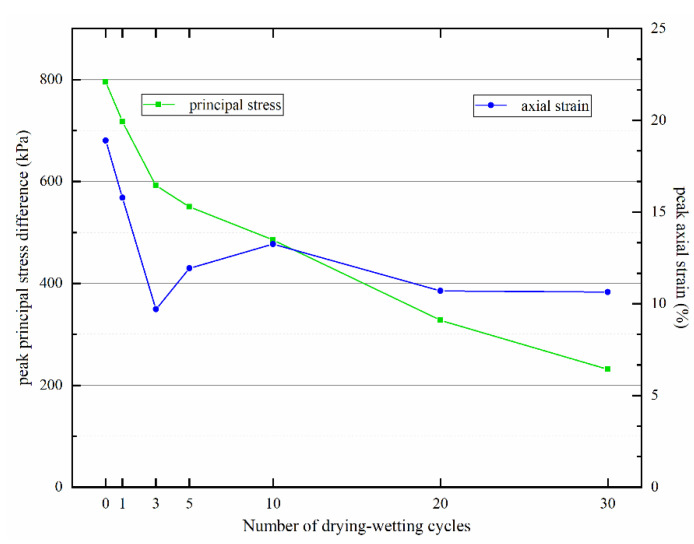
Variations of characteristic parameters of stress–strain curves with the number of drying–wetting cycles.

**Figure 7 materials-15-00255-f007:**
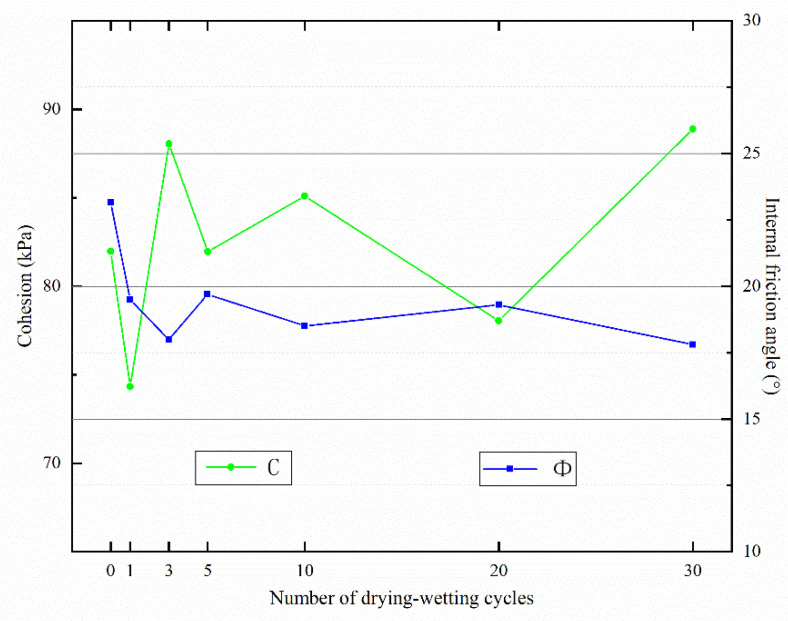
Variation of shear strength parameters with the number of drying–wetting cycles.

**Figure 8 materials-15-00255-f008:**
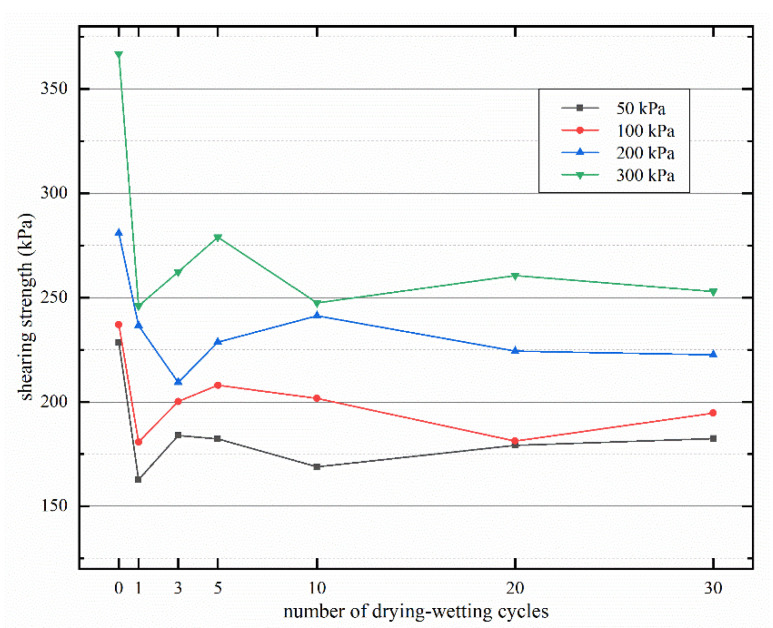
Variation of shear strength with the number of drying–wetting cycles.

**Figure 9 materials-15-00255-f009:**
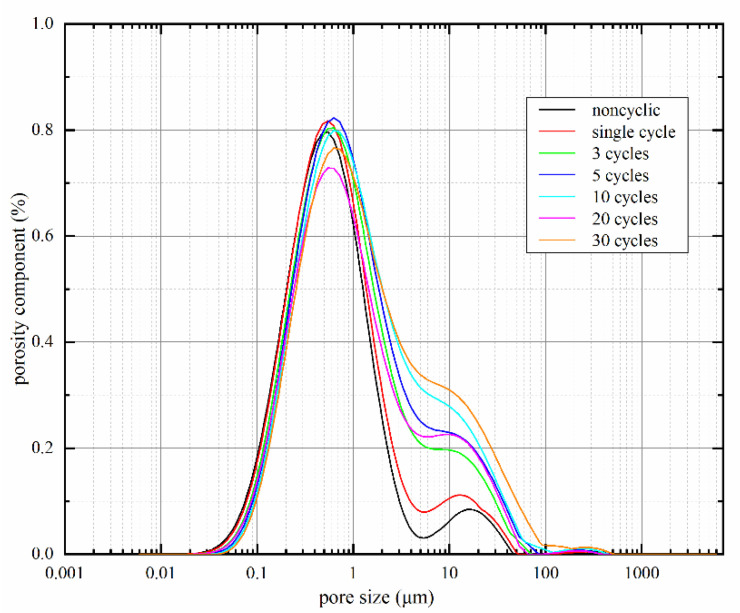
Pore size distribution patterns of Yili loess samples after different numbers of drying–wetting cycles.

**Figure 10 materials-15-00255-f010:**
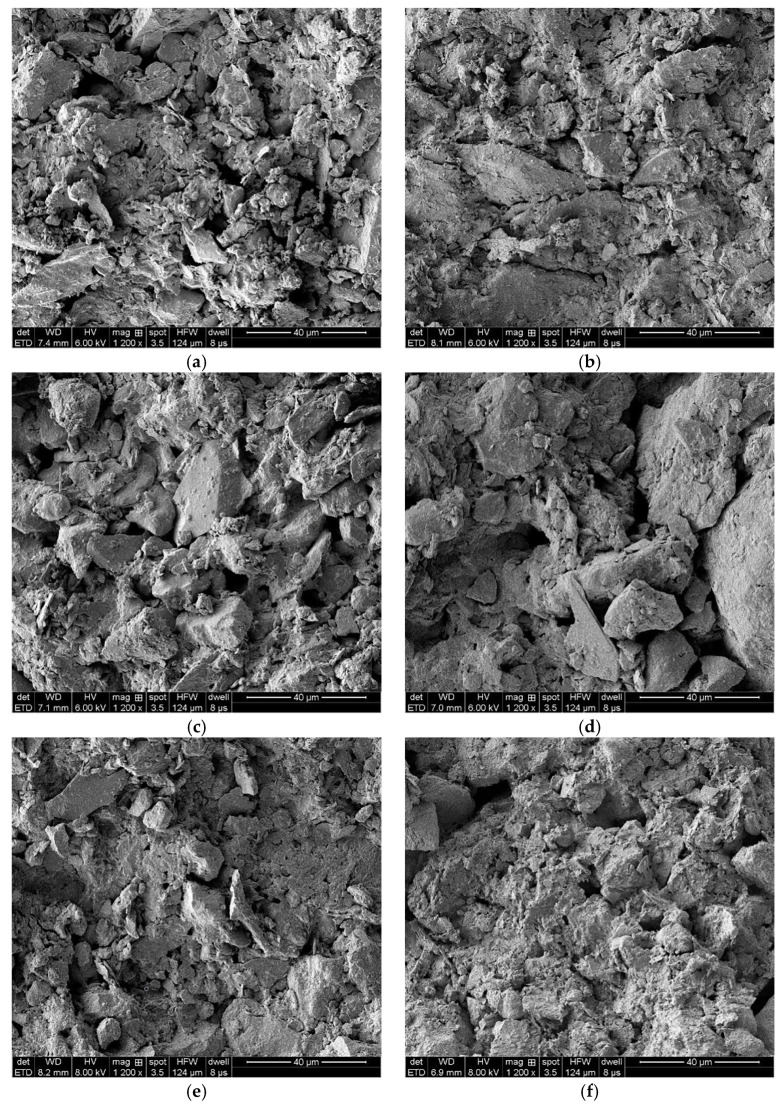
SEM images of Yili loess samples after different drying–wetting cycles (**a**) 0, (**b**) 1, (c) 3, (**d**) 5, (**e**) 10, (**f**) 20, (**g**) 30.

**Figure 11 materials-15-00255-f011:**
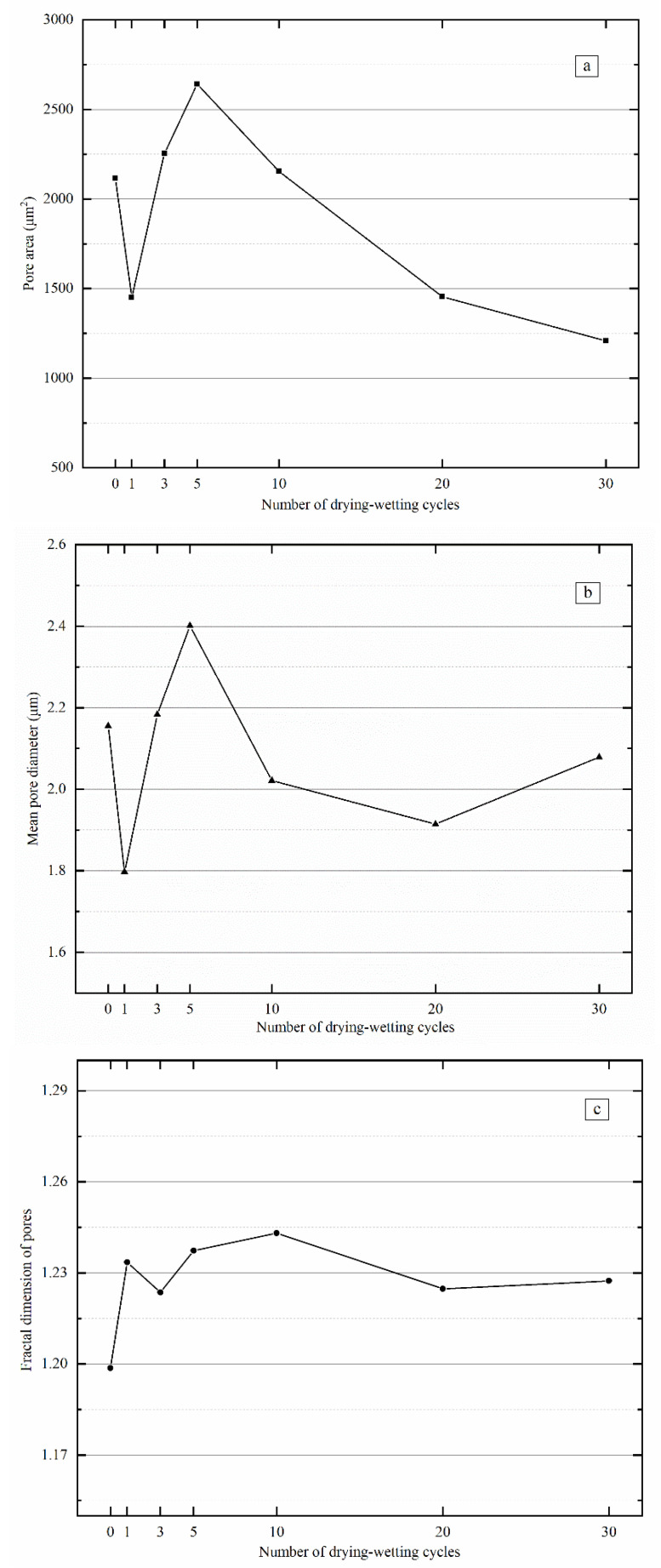
Relationship between the number of drying–wetting cycles and microstructural parameters. (**a**) Pore Area, (**b**) Mean Pore Diameter, (**c**) Fractal Dimension of Pores.

**Table 1 materials-15-00255-t001:** Basic physical properties.

Natural Moisture Content (%)	Saturated Moisture Content (%)	Optimum Moisture Content (%)	Maximum Dry Density (g/cm^3^)	Porosity	Void Ratio
20.89	24.57	17.4	1.86	28.34	22.08%

**Table 2 materials-15-00255-t002:** Mineral component analysis results.

Quartz	Calcite	Dolomite	Clinochlore llb	Albite	Muscovite 2M	Rutile
28.1	21.1	3.2	10.5	19.5	15	1.5

**Table 3 materials-15-00255-t003:** Soluble salt content of Yili loess.

	Soluble Salt Content (g/kg)	Ion Content (g/kg)							
		Na^+^	K^+^	Mg^2+^	Ca^2+^	CO_3_^2−^	HCO_3_^−^	Cl^−^	SO_4_^2−^
Lisi Niu et al.	3.54	1.645	0	0.073	0.098	0	0.287	0.206	1.228
Wang Yu-guo et al.	4.9361	0.9532	0	0.0287	0.0246	0.2795	2.8419	0.1663	0.9285

**Table 4 materials-15-00255-t004:** Numbers of the loess samples in different tests.

Number of Drying–Wetting Cycles	Number of Loess Samples for Tri-Axial Shear Test	Number of Loess Samples for Microstructural Analysis	
		Nuclear Magnetic Resonance Test (NMR)	Scanning Electron Microscopy Test (SEM)
0	4	1	1
1	4	1	1
3	4	1	1
5	4	1	1
10	4	1	1
20	4	1	1
30	4	1	1

**Table 5 materials-15-00255-t005:** Shear strength parameters.

Number of Drying–Wetting Cycles	Water Content (%)	Internal Friction Angle (°)	Cohesion (kPa)
0	17.4	23.1575	81.97
1	17.4	19.4994	74.34
3	17.4	17.9911	88.06
5	17.4	19.6997	81.95
10	17.4	18.5069	85.09
20	17.4	19.3031	78.06
30	17.4	17.8061	88.89

**Table 6 materials-15-00255-t006:** Shear strength.

Number of Drying–Wetting Cycles	Shear Strength (kPa)			
	σ_3_ = 50	σ_3_ = 100	σ_3_ = 200	σ_3_ = 300
0	228.51	237.11	280.86	366.83
1	162.73	180.78	236.52	245.98
3	183.96	200.20	209.35	262.25
5	182.24	208.02	228.68	278.95
10	168.84	201.71	241.21	247.53
20	179.21	181.25	224.40	260.58
30	182.48	194.66	222.63	252.95

**Table 7 materials-15-00255-t007:** Porosities of Yili loess samples after different numbers of drying–wetting cycles.

Number of Drying–Wetting Cycles	Porosity
0	28.34
1	30.55
3	34.59
5	36.60
10	37.82
20	32.96
30	38.56

**Table 8 materials-15-00255-t008:** Microstructural parameters of Yili loess samples after different drying–wetting cycles.

Number of Drying–Wetting Cycles	Pore Area	Pore Diameter			Fractal Dimension of Pores
		Maximum	Minimum	Mean	
	(μm^2^)	(μm)			
0	2114.5757	13.4953	0.6330	2.1552	1.1986
1	1449.0347	12.8381	0.3903	1.7972	1.2335
3	2254.8845	12.4943	0.5985	2.1832	1.2236
5	2641.0435	31.6204	0.7420	2.4009	1.2374
10	2154.9717	16.7605	0.5561	2.0213	1.2431
20	1455.1604	14.0038	0.6631	1.9142	1.2248
30	1207.9547	10.6068	0.6523	2.0785	1.2274

**Table 9 materials-15-00255-t009:** Original data of microstructure and shear strength parameters under cyclic drying–wetting action.

Number of Drying–Wetting Cycles	Mean Pore Diameter (μm)	Fractal Dimension of Pores	Internal Friction Angle (°)	Cohesion
0	2.1552	1.1986	23.1575	81.97
1	1.7972	1.2335	19.4994	74.34
3	2.1832	1.2236	17.9911	88.06
5	2.4009	1.2374	19.6997	81.95
10	2.0213	1.2431	18.5069	85.09
20	1.9142	1.2248	19.3031	78.06
30	2.0785	1.2274	17.8061	88.89

**Table 10 materials-15-00255-t010:** Dimensionless processing results.

Number of Drying–Wetting Cycles	Mean Pore Diameter	Fractal Dimension of Pores	Internal Friction Angle	Cohesion
0	1.0368	0.9769	1.1922	0.9921
1	0.8646	1.0054	1.0039	0.8998
3	1.0503	0.9973	0.9263	1.0658
5	1.1550	1.0085	1.0142	0.9919
10	0.9724	1.0132	0.9528	1.0299
20	0.9209	0.9983	0.9938	0.9448
30	0.9999	1.0004	0.9167	1.0759

**Table 11 materials-15-00255-t011:** Calculated correlation coefficients between shear strength parameters and microstructural parameters of Yili loess samples under drying–wetting cyclic action.

Number of Drying–Wetting Cycles	Internal Friction Angle		Cohesion	
	Mean Pore Diameter	Fractal Dimension of Pores	Mean Pore Diameter	Fractal Dimension of Pores
0	0.415	0.338	0.766	1.000
1	0.442	1.000	0.829	0.517
3	0.471	0.611	0.997	0.645
5	0.439	0.963	0.395	0.985
10	0.857	0.649	0.696	0.985
20	0.604	0.973	0.917	0.716
30	0.572	0.570	0.614	0.616

**Table 12 materials-15-00255-t012:** Calculated correlation degrees between shear strength indices and microstructural parameters of Yili loess samples under drying–wetting cyclic action.

Internal Friction Angle		Cohesion	
Mean Pore Diameter	Fractal Dimension of Pores	Mean Pore Diameter	Fractal Dimension of Pores
0.543	0.729	0.745	0.780

## Data Availability

The data used to support the findings of this study are included within the manuscript.

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
