# Peer review of "Investigation of Changes to Triaxial Shear Strength Parameters and Microstructure of Yili Loess with Drying–Wetting Cycles"

_materials, 2021, doi:10.3390/ma15010255_

Round 1
Reviewer 1 Report
The manuscript ‘Investigation of changes to triaxial shear strength parameters and microstructure of Yili loess with drying-wetting cycles’ is very well written and elaborated. The topic is also interesting and falls in the scope of the journal.
Some additional explanations are required prior to publishing the manuscript:
- Figure 1 needs to be changed with higher resolution figure
- Figure 7 – also increase quality
- Figure 11 – the same
- Figure 13 is not useful
- Why do you crush the sample into the particles less than 2 mm. This way you change its microstructure and corresponding macro properties. This is not the same material like original that you want to obtain properties for. Can you please explain this point? Check the reference below.
- The statements in the abstract and in the manuscript (page 7 line 178) are contradictory. The statement in abstract is that internal friction increases with drying-wettig cycles, and in the text is the opposite. Please explain/correct
- How is Yili loess behaviour different from the marl during drying wetting cycles presented in the following work
P Miščević, G Vlastelica, M Nikolić, LABORATORY INVESTIGATION OF EMBANKMENT SETTLEMENT CAUSED BY MARL GRAINS DETERIORATION, Gradevinski fakultet, University of Mostar 8 (16), 68-75, https://hrcak.srce.hr/clanak/312972
Reviewer 2 Report
The research idea deserves attention, proposed approaches are worth consideration. Unfortunately, the quality of the scientific overall presentation is very poor. The major issue here is that the authors try to make conclusions based on sem analyses of a single sample (for each wet-dry cycle), from scientific evidence perspective that is not representative at all. The methods and the materials are barely explained nor the crucial details on how the parameters were measured are presented, not even the sample preparation methods were revealed. It's not clear how the authors managed to perform 30 wetting cycles without damaging the soil sample. The cleaning film used in the dehumidification process would not allow the change of water content. All these are very unclear. The introduction part does not provide any info (nor the abstract) on the aim of the study, what is the purpose of analyses this particular soil material in terms of hydro-mechanic behaviour apply cycling stresses. The soil material should be described in more detail, please review the available literature to provide information on elastic, hydraulic (k) and mechanical conditions in standard tests as a reference. The big question that needs to be answered is why the authors decided to analyse this type of material? and why determining mechanical parameters in cyclic stress conditions is important to establish? When it comes to methodology attention to detail is needed. More critical reviews on the sample preparation process need to be added. How the loess sample was protected from dissolving in the water, after the 30th wetting cycle? What about the changing soil structure, and sample shape so it is suitable for trx tests? Also drying process needs more attention... how was the cracking avoided? What about the volume change of the sample? if the samples were kept in cleaning film the how dehumidification was allowed? The analysis of results needs significant improvement. The results for shear strength are not stabilising at all. The graph shows dropping-increasing-dropping-increasing behaviour. Obviously, the first cycle would give lower values due to wetting. Please refer that to other researches. To some extent the manuscript focuses on statistical analyses, however, making performing it for a single sample and presenting it as scientific evidence is not convincing in this case. Please consider testing more samples or simply exclude this part from analyses. The Discussion needs a lot more critical review, bearing in mind all the remarks given in all sections. These are only overall and major comments. More specific comments are that the quality of Figures No. 1, 3, 4, 7-11 and 14 must be improved. The References should be expanded and updated to more recent studies. Another thing is all of them are Asia research-oriented, to make the literature review broader and representative is suggested to review other researchers work too. Mandatory change is style and grammar. A number of sentences are difficult to follow. Below the authors will find more specific comments
Specific comments:
l.41 why is this information relevant in the introduction? this is simply describing the testing material origin with too many details
l.47 Please rewrite the sentence, it is obvious that the moisture content is affecting moisture characteristics. This sentence does not make much sense. Do the scholars believe? I want to 'believe' that the researchers possess scientific evidence to make such a statement.
l.61 soil pores change or porosity? how the particles can change during the wetting-drying process. The sentence is not clear.
tab 2 since wetting and drying is changing the soil suction (influencing the shear strength) information on salinity is needed here too.
l.90 was the water distribution measured in each section of samples, or was it only assumed?
l.104 how was it measured, was it confirmed anyhow that moisture content is reached?
l.130 why uu test was selected, this needs clear justification (practical and theoretical implications)
l.143. does it mean that both, sheared samples and samples before the shearing were scanned? please make sure you are precise in providing information on testing procedures.
l.221 Pattern cannot be referred to when analysing a single sample
The specific comments are only a few remarks that need addressing in the main text. Please see the attached copy where more comments could be followed.

Round 2
Reviewer 1 Report
Authors responded correctly to my comments. The manuscript can be accepted.
Reviewer 2 Report
The authors introduced corrections and supplements in accordance with the comments of the reviewers. Still minor explanations can be introduced in the descriptions of the research design, the methods used and the test results. The article can also be accepted for publication in its present form. Good luck!